# Periselectivity in the Aza-Diels–Alder Reaction of 1-Azadienes with α-Oxoketenes: A Combined Experimental and Theoretical Study

**DOI:** 10.3390/molecules25204811

**Published:** 2020-10-20

**Authors:** Marc Presset, Michel Rajzmann, Guillaume Dauvergne, Jean Rodriguez, Yoann Coquerel

**Affiliations:** Aix Marseille Université, CNRS, Centrale Marseille, ISM2, 13397 Marseille, France; presset@icmpe.cnrs.fr (M.P.); michel.rajzmann@univ-amu.fr (M.R.); guillaume.dauvergne@univ-amu.fr (G.D.)

**Keywords:** hetero-Diels–Alder, ketenes, imines, periselectivity, density functional theory

## Abstract

Inversions in the periselectivity of formal aza-Diels–Alder cycloadditions between α-oxoketenes generated by a thermally-induced Wolff rearrangement and 1-azadienes were observed experimentally as a function of the α-oxoketene and the 1-azadiene, as well as the reaction temperature and time. Some unexpected inversion in the diastereoselectivity was observed, too. These variations in selectivities were fully rationalized by computational modeling using density functional theory (DFT) methods.

## 1. Introduction

α-Oxoketenes [1,2,3,4,5,6] were first employed in organic synthesis in 1902 with the pioneering work of Ludwig Wolff on 2-diazo-1,3-dicarbonyl compounds [7]. Since, α-oxoketenes have revealed as versatile chemical intermediates for synthetic applications, ranging from the preparation of simple β-oxocarbonyl derivatives [8] to the total synthesis of complex natural products [4,5]. However, and with few exceptions [9,10], α-oxoketenes are not stable species because of their propensity to react with a number of chemicals, including themselves [11], at significant rates at ambient temperature; thus, they must be generated in situ immediately before use. The most common and practical methods for the generation of α-oxoketenes are the thermal decomposition of dioxinones with concomitant extrusion of acetone [12], and the thermal or photochemical Wolff rearrangement of 2-diazo-1,3-dicarbonyl compounds extruding nitrogen gas [13,14]. α-Oxoketenes **1** are electrophilic species that most commonly react with imines through formal aza-Diels–Alder cycloadditions involving the α-oxoketenes as 4π reaction partners (1-oxadienes) and the imines as the 2π reaction partners to give 1,3-oxazin-4-ones of type **4** (Scheme 1) [1,2,3,4,5,6,15]. However, when α,β-unsaturated imines **2** are employed, another formal aza-Diels–Alder process becomes possible, involving the α,β-unsaturated imines as 4π reaction partners (1-azadienes) and the α-oxoketenes as the 2π reaction partners to give hydropyridin-2-ones of type **3** [16]. Periselectivity, the selectivity in formation of the pericyclic products, in the formal aza-Diels–Alder cycloadditions between α-oxoketenes and some electron rich 2-azadienes was recently examined and found to be dependent on kinetic and thermodynamic factors, as well as the nature of the α-oxoketene employed [11]. For the present study, the periselectivity in the reactions of α-oxoketenes with α,β-unsaturated imines as prototypical 1-azadienes was further investigated and rationalized by a combination of experimental and computational studies.

## 2. Results and Discussion

We previously reported the formal aza-Diels–Alder cycloaddition between the α,β-unsaturated imine **2a** and the five-membered cyclic α-oxoketene **1a** derived from the thermal Wolff rearrangement of 2-diazodimedone to give the spiro hydropyridin-2-one **3a** in 24% yield after 15 min of reaction at 160 °C (Scheme 2a) [16]. Re-examination of the temperature dependence of the reaction outcome revealed that the other pericyclic product, namely the 1,3-oxazin-4-one **4a**, is actually formed when the reaction is performed at 140 °C with a 2:1 **3a**/**4a** ratio after 15 min, which allowed isolating **3a** in 38% yield and **4a** in 21% yield. A similar reaction conducted at 150 °C for 30 min afforded only **3a** in 22% isolated yield (**3a**/**4a** > 25:1). Stimulated by these observations, some complementary experiments were performed to further illustrate the possible change in periselectivity in the aza-Diels–Alder cycloadditions between α-oxoketenes and 1-azadienes as a function of the reaction temperature (Scheme 2b–e). For instance, the reaction between α-oxoketene **1a** and the enantiopure 1-azadiene **2b** derived from (–)-perillaldehyde and allylamine at 140 °C for 15 min afforded the corresponding diastereomeric 1,3-oxazin-4-ones **4b** in 59% (dr = 1:1) while it was previously shown to afford the spiro hydropyridin-2-one **3b** after 15 min at 200 °C [16]. Similarly, it was found earlier that spiro hydropyridin-2-one **3c** could be obtained efficiently from the 2-methyl-1-azadiene **2c** from a reaction at 160 °C [16], while a mixture of the two possible pericyclic products **3c** and **4c** was obtained at 140 °C. Some degree of thermodynamic control of the reaction could be demonstrated experimentally in this case: heating a toluene solution of **4c** at 160 °C for 15 min produced 25% of **3c** together with 20% of 2-methylcinnamaldehyde and unidentified products. Surprisingly, the reactions of α-oxoketene **1a** with the 3-chloro 1-azadiene **2d**, led exclusively to the corresponding 1,3-oxazin-4-one **4d** with no detectable amount of **3d**, at 140 °C or 200 °C. The influence of the nature (phenyl vs. methyl) of the substituent at position 4 of the 1-azadiene on periselectivity was briefly evaluated through the reaction of α-oxoketene **1a** with 1-azadienes **2e** (R = Ph) and **2f** (R = Me) at 140 °C for 15 min. It was earlier found that the reaction with 1-azadiene **2e** provided the spiro hydropyridin-2-one **3e** in 70% yield [16]. The same reaction conducted with azadiene **2f** provided only the 1,3-oxazin-4-one **4f** albeit in low yield. 

The influence of the nature of the α-oxoketene on periselectivity was also evaluated experimentally. The six-membered cyclic α-oxoketene **1b** derived from the thermal Wolff rearrangement of 2-diazocycloheptan-1,3-dione was reacted with 1-azadiene **2g** at 160 °C to afford cleanly the 1,3-oxazin-4-one **4g**, while the same reaction mixture at 200 °C for a prolonged time previously afforded the spiro hydropyridin-2-one **3g** in 53% yield (Scheme 3a) [16]. The aza-Diels–Alder reaction of the acyclic α-oxoketene **1c** with the 1-azadiene **2h** was also examined and found to lead to the 1,3-oxazin-4-one **4h** when conducted at 140 °C for 5 min, while the same reaction performed at 200 °C for 15 min resulted in the exclusive formation of **iso-3h** (Scheme 3b). Significantly, the relative stereochemistry in **iso-3h** is opposite to the stereochemistry that has been described so far in this and other series of cycloadditions of α-oxoketenes with 4π reaction partners [11,16,17]. Ultimately, it was shown that α-oxoketene **1d** derived from the Wolff rearrangement of methyl 2-diazoacetylacetate reacted with the 1-azadiene **2i** at 160 °C to give the 1,3-oxazin-4-one **4i** in 43% yield, and no reaction at higher temperature was attempted.

These new experimental data indicate that periselectivity in the aza-Diels–Alder cycloadditions between α-oxoketenes and 1-azadienes is generally controlled by competitive kinetic and thermodynamic factors. However, there are some marked differences in reactivity within relatively homogeneous series of substrates, which called for rationalization to enable further developments and applications. To do so, the problem was tackled through computational modeling using density functional theory (DFT) methods. The α-oxoketenes **1b,c** and **1e** and the model 1-azadienes **2j–m** were selected for this part of the work. All calculations were performed using the B3LYP-D3/6-311++G(d,p) level of theory. Free Gibbs energies were generally computed at 298 K corresponding to the temperature of isolation of the products, and additionally at the reaction temperature for kinetically meaningful steps (see the Appendix A for details of the calculations).

First, we examined the reactions between the α-oxoketene **1e** and the 1-azadiene **2j** to afford either the spiro hydropyridin-2-one **3j** or the 1,3-oxazin-4-one **4j** (Figure 1a,b). As expected, formation of **3j** was found largely thermodynamically favored when compared to the formation of **4j** [Δ(ΔG) = −56.1 kJ/mol] with barriers computed at Δ(ΔG^≠^) = 12.1 kJ/mol at 298 K and 18.9 kJ/mol at 413 K, which is fully consistent with the experimental results in Scheme 2a. However, the actual aza-Diels–Alder thermodynamic product of the reaction was computationally identified as **iso-3j** (the diastereomer of **3j**) with a relative free Gibbs energy determined at −75.8 kJ/mol, that is more stable than **3j** by a tiny but meaningful −2.4 kJ/mol; modeling its formation indicated a relative activation energy of ΔG^≠^ = 73.9 kJ/mol at 298 K and 100.5 kJ/mol at 413 K plausible with its formation under the actual reaction conditions (Figure 1c). Because the diastereoselectivity in the model molecule **iso-3j** has never been reported experimentally in cycloadditions of α-oxoketene **1e** with any 4π reaction partner [11,16,17], a competing process is certainly at hand. Actually, the formation of **iso-3j** through **TS6** was found kinetically outcompeted by the cyclodimerization of α-oxoketene **1e** (Figure 1c) that occurs with a significantly lower relative activation energy through **TS7** to give **dimer-1e**, a molecule itself amenable to further thermodynamically driven transformations under the reaction conditions [11]. 

The aza-Diels–Alder reactions of α-oxoketene **1e** with the 3-substituted 1-azadienes **2k** (Figure 2) and **2l** (Figure 3) were next examined in silico for comparison with the experimental results in Scheme 2c,d. As above, formation of the 1,3-oxazin-4-ones **4k** and **4l** was found kinetically preferred to the formation of the corresponding spiro hydropyridin-2-ones of **3k** and **3l**, with two competing reaction paths identified for the formation of **4l** (Figure 3a). This is in agreement with the experimentally observed formation of both 1,3-oxazin-4-ones **4c** and **4d** from reactions at 140 °C. However, at 160 °C, the methyl-substituted spiro hydropyridin-2-one **3c** (Scheme 2c) corresponding to the model molecule **3k** (Figure 2b) was obtained experimentally, while somehow unexpectedly the chloro-substituted spiro hydropyridin-2-one **3d** (Scheme 2d) corresponding to the model molecule **3l** (Figure 3b) could not be obtained even at 200 °C. The rationalization of this experimental fact came from the comparison of the relative activation energies for the formation of **3k**, **3l**, and **dimer-1e** at 200 °C: **TS11** leading to **3k** competes favorably with the formation of **dimer-1e** through **TS7**, though not by much with Δ(ΔG^≠^) = −3.5 kJ/mol at 473 K, while **TS17** leading to **3l** competes unfavorably with the formation of **dimer-1e** with Δ(ΔG^≠^) = 3.4 kJ/mol at 473 K. 

To conclude with the modeling of reactions of the cyclic five-membered α-oxoketene **1e** with 1-azadienes, the influence of a 4-phenyl or a 4-methyl substituent on the 1-azadiene was investigated to rationalize the results in Scheme 2e. Comparison of the energy profiles of the reactions α-oxoketene **1e** with the model 1-azadienes **2j** (Figure 1) and **2m** (Figure 4) essentially shows that the formation of the model spiro hydropyridin-2-one **3m** bearing a methyl substituent is significantly slower than the reaction leading to the model spiro hydropyridin-2-one **3j** bearing a phenyl substituent with Δ(ΔG^≠^) = 12.4 kJ/mol at 298 K and 34.8 kJ/mol at 413 K (compare **TS4** with **TS21**). The relative increase in the activation energy induced by the 4-methyl substituent is significant and precludes the formation of the corresponding spiro hydropyridin-2-one **3m** at the benefit of the α-oxoketene cyclodimer **dimer-1e** for kinetic reasons. Altogether, the selected DFT model allowed a faultless rational of the periselectivity in the reactions of α-oxoketene **1e** with 1-azadienes, which was found to be governed by a combination of kinetic and thermodynamic factors.

Calculations with the cyclic six-membered α-oxoketene **1b** (Figure 5) paralleled the previous findings and were also found in full agreement with the experimental results, with the actual thermodynamic diastereomer of the spiro hydropyridin-2-one product being computationally identified as **iso-3n**, that is not the model molecule corresponding to the experimentally observed diastereomer **3g** from the reaction at 200 °C (Scheme 3a). At 200 °C, there are two orders of magnitude between the kinetic constants estimated for the formation of **3n** via **TS25** and **iso-3n** via **TS27** (circa k = 60 s^−1^ and 0.56 s^−1^, respectively), but the formation of **iso-3n** remains plausible after 45 min at 200 °C. It should, however, be considered that the ratio **Int10**/**Int11** is higher than 10^4^:1 according to the model, which precludes the formation of **iso-3n** at a significant rate under the examined experimental conditions (relative rates in the magnitude of 10^6^:1). Additionally, the cyclodimerization of α-oxoketene **1b** to give **dimer-1b** through **TS28** is also in competition with the formation of **iso-3n** at 200 °C. On this occasion, again, the selected DFT model allowed for a complete rationalization of the periselectivity in the reactions of α-oxoketene **1b** with 1-azadienes, which was found to be governed by a combination of kinetic and thermodynamic factors, as for α-oxoketene **1e**.

The calculations in the acyclic series were complicated by the existence of the additional *s-trans* conformer of the α-oxoketene **1c** (Figure 6a) but again were in full agreement with the experiments. As expected, **4o** was identified as the kinetic aza-Diels–Alder product in the model reaction between α-oxoketene **1c** and 1-azadiene **2j** with ΔG^≠^ = 70.7 kJ/mol at 298 K and 102.5 kJ/mol at 473 K. As above, the true thermodynamic cycloadduct was identified as **iso-3o** with notably a more pronounced difference in stabilization energy between the two diastereomers **3o**/**iso-3o** than in the previous cases for **3j**/**iso-3j** (Figure 1) and **3n**/**iso-3n** (Figure 5). In addition, the difference between the activation energies leading to one or the other diastereomer is reduced in this series with Δ(ΔG^≠^) = 10.3 kJ/mol at 298 K and 10.9 kJ/mol at 473 K for **TS33**/**TS35**. This is to be compared with Δ(ΔG^≠^) = 17.0 kJ/mol at 298 K and 17.1 kJ/mol at 473 K for **TS4**/**TS6**, and Δ(ΔG^≠^) = 18.8 kJ/mol at 298 K and 18.4 kJ/mol at 473 K for **TS25**/**TS27**. The difference between the estimated kinetic constants corresponding to **TS33** and **TS35** is now reduced to a single order of magnitude (circa k = 0.49 s^−1^ and 0.031 s^−1^ for **TS33** and **TS35** at 200 °C, respectively), allowing for both processes to compete at 200 °C. In this case, the cyclodimerization of the α-oxoketene **1c** (via **TS36**) was found not to compete kinetically with the aza-Diels–Alder processes examined. Altogether, the calculations indicate that in the case of the acyclic α-oxoketene **1c**, and only in this case, the actual thermodynamic product of the reaction **iso-3o** can be obtained, which was demonstrated experimentally with the isolation of **iso-3h** (Scheme 3b). 

All the aza-Diels–Alder processes described herein are stepwise processes, occurring first with the formation of the N–C bond to produce a zwitterionic intermediate and then cyclization to the product with formation of the C–C or O–C bond. These processes should thus preferably be referred to as *formal* aza-Diels–Alder cycloadditions. The physical reason at the origin of this behavior is that the LUMO of all the α-oxoketenes are essentially located over the ketene carbonyl groups, not over the two or four sp^2^ hybridized atoms involved in the cycloaddition, and directed in the mean plane of the α-oxoketenes, not perpendicular to it as in standard Diels–Alder cycloadditions (Figure 7). Actually, the LUMO+1 of the α-oxoketenes are mostly distributed over the α-oxo carbonyl groups and the ketene C=C bonds and oriented perpendicular to the molecular plane [11], but with a significant LUMO/LUMO+1 gap of 0.01453 Ha or 38.1 kJ/mol on average, and no transition states accounting for the symmetry-allowed concerted transformations were located computationally during this work.

In the course of the computational investigations summarized herein, some alternative kinetically, and for the most also thermodynamically, disfavored reaction paths leading to other cycloadducts were identified; this includes β-lactams resulting from formal [2+2] cycloadditions between the C=C bond of the ketene and the C=N bond of the 1-azadiene, and 1,3-oxazin-2-ylidenes resulting from a formal [4+2] cycloaddition between the C=O bond of the ketene and the 4π system of the 1-azadiene.

## 3. Experimental Section

All reagents were purchased from commercial sources and used without further purification unless otherwise noted. All compounds were weighed and handled in air at room temperature. Microwave-assisted heating was performed using a professional Anton-Paar (Austria) Monowave 300 system in specific sealed tubular reaction vessels. The reactions were monitored by thin layer chromatography (TLC, 60 F_254_) visualized by UV lamp (254 nm or 365 nm). Flash chromatography was performed on silica gel (particle size 40−63 µm). Petroleum ether refers to the fraction with bp = 40–60 °C. NMR spectra were recorded at 300 MHz (^1^H) and 75 MHz (^13^C) at 298 K in CDCl_3_ using as internal standards the residual non-deuterated signal for ^1^H-NMR (δ = 7.26 ppm) and the deuterated solvent signal for ^13^C-NMR spectroscopy (δ = 77.16 ppm). DEPT-135 experiments were used to determine the multiplicity of the ^13^C resonances. Chemical shifts (δ) are given in ppm, coupling constants (*J*) are given in Hz, and the classical abbreviations are used to describe the multiplicity of the ^1^H resonances. Copies of all NMR spectra are provided as Appendix A for this article. High-resolution mass spectra were recorded in triplicate at the Spectropole (https://fr-chimie.univ-amu.fr/spectropole/). 2-Diazo-1,3-dicarbonyls were prepared as previously reported in similar yields and purity [18].

General procedure for the aza-Diels–Alder reactions: a solution of a primary amine (1 equiv) and an α,β-unsaturated aldehyde (1 equiv) in 2–3 mL of anhydrous toluene (ca. 0.4 M) under an argon atmosphere in a microwave dedicated sealed tube containing a Teflon coated magnetic stirring bar was irradiated at 140 °C for 15 min (ramp up time = 2 min), cooled down to 55 °C by an air flow over 5−6 min, and concentrated directly in the reaction vessel to afford the corresponding imine quantitatively as verified by ^1^H-NMR analysis. To this material placed under an argon atmosphere was added the diazo compound (1 equiv) and anhydrous toluene (2–3 mL, ca. 0.4 M), the reaction vessel was sealed, and this mixture was irradiated at the set temperature for the time indicated (after a temperature ramp-up time of ca. 2 min). The resulting reaction mixture was cooled down to 55 °C by an air flow over 5−6 min, concentrated, analyzed by NMR to determine the products ratio, and directly purified by flash chromatography when isolation was attempted. 

**4a**: Following the general procedure, the reaction between *ortho*-nitrocinnamaldehyde (172 mg, 0.97 mmol), furfurylamine (86 μL, 0.97 mmol), and 5,5-dimethyl-2-diazo-cyclohexan-1,3-dione (161 mg, 0.97 mmol) at 140 °C for 15 min afforded the spiro hydropyridin-2-one **3a** (147 mg, 38%) as a yellow oil and the 1,3-oxazin-4-one **4a** (80 mg, 21%) as a yellow oil. Rf (AcOEt/petroleum ether 4:6) = 0.45. HRMS (ESI+) *m/z*: [M+H]^+^ calcd for C_22_H_23_N_2_O_5_^+^ 395.1601, found 395.1604. ^13^C{^1^H}-NMR (75 MHz, δ ppm/CDCl_3_): 165.8 (C), 161.8 (C), 150.4 (C), 147.8 (C), 142.3 (CH), 133.4 (CH), 131.3 (C), 131.0 (CH), 129.1 (CH), 128.9 (CH), 127.2 (CH), 124.7 (CH), 110.5 (CH), 108.8 (CH), 108.3 (C), 88.7 (CH), 46.2 (CH_2_), 41.0 (CH_2_), 38.6 (CH_2_), 36.3 (C), 30.0 (CH_3_), 29.9 (CH_3_). ^1^H-NMR (300 MHz, δ ppm/CDCl_3_): 7.99 (dd, *J* = 1.6, 7.8 Hz, 1H), 7.59 (ddd, *J* = 1.6, 7.4, 7.4 Hz, 1H), 7.50–7.43 (m, 2H), 7.33 (dd, *J* = 0.9, 1.5 Hz, 1H), 7.15 (d, *J* = 15.7 Hz, 1H), 6.32–6.23 (m, 3H), 5.85 (dd, *J* = 0.9, 6.7 Hz, 1H), 4.90 (d, *J* = 15.8 Hz, 1H), 4.40 (d, *J* = 15.8 Hz, 1H), 2.46–2.25 (m, 4H), 1.15 (s, 3H), 1.13 (s, 3H). 

**4b**: Following the general procedure, the reaction between (–)-perillaldehyde (155 μL, 1.0 mmol), allylamine (75 μL, 1.0 mmol), and 5,5-dimethyl-2-diazo-cyclohexan-1,3-dione (166 mg, 1.0 mmol) at 140 °C for 15 min afforded the 1,3-oxazin-4-one **4b** (193 mg, 59%, dr = 1:1) as a yellow oil. Rf (Et_2_O/petroleum ether 2:8) = 0.18. HRMS (ESI+) *m/z*: [M+H]^+^ calcd for C_21_H_30_NO_2_^+^ 328.2271, found 328.2271. ^13^C{^1^H}-NMR (75 MHz, δ ppm/CDCl_3_): 165.3 (C), 164.9 (C), 163.0 (C), 162.7 (C), 148.8 (C), 148.6 (C), 133.3 (CH), 133.3 (CH), 132.0 (C), 131.8 (C), 128.9 (CH), 128.2 (CH), 117.0 (CH_2_), 116.9 (CH_2_), 109.0 (CH_2_), 109.0 (CH_2_), 107.9 (C), 107.7 (C), 91.7 (CH), 91.3 (CH), 46.0 (CH_2_), 46.0 (CH_2_), 44.8 (CH_2_), 44.6 (CH_2_), 41.0 (CH_2_), 40.9 (CH_2_), 40.6 (CH), 40.2 (CH), 36.1 (C), 36.0 (C), 30.2 (CH_2_), 30.0 (CH_2_), 29.9 (CH_3_), 29.9 (CH_3_), 29.8 (CH_3_), 29.7 (CH_3_), 27.0 (CH_2_), 26.7 (CH_2_), 24.6 (CH_2_), 23.6 (CH_2_), 20.7 (CH_3_), 20.6 (CH_3_). ^1^H-NMR (300 MHz, δ ppm/CDCl_3_): 5.79–5.66 (m, 4H), 5.46 (br, 2H), 5.15–5.07 (m, 4H), 4.10–4.65 (m, 4H), 4.59–4.45 (m, 2H), 3.31–3.20 (m, 2H), 2.32–2.25 (m, 8H), 2.16–1.90 (m, 8H), 1.86–1.80 (m, 2H), 1.69 (s, 6H), 1.50–1.36 (m, 2H), 1.21–1.14 (m, 2H), 1.10 (s, 6H), 1.08 (s, 6H). 

**4c**: Following the general procedure, the reaction between α-methyl-cinnamaldehyde (140 μL, 1.0 mmol), benzylamine (109 μL, 1.0 mmol), and 5,5-dimethyl-2-diazo-cyclohexan-1,3-dione (167 mg, 1.0 mmol) at 140 °C for 15 min afforded a 2:1 mixture of the 1,3-oxazin-4-one **3c** and the 1,3-oxazin-4-one **4c** (311 mg, 82% total) that we failed at separating by flash chromatography. Some amount of 1,3-oxazin-4-one **4c** could be isolated by treatment of the **3c**/**4c** 2:1 mixture by 0.75 equiv NaBH_4_ in ethanol followed by standard work-up and flash chromatography eluted with Et_2_O/petroleum ether 2:8 because **4c** is chemically inert under these conditions. Rf (Et_2_O/PE 2:8) = 0.21. HRMS (ESI+) *m/z*: [M+H]^+^ calcd for C_25_H_28_NO_2_^+^ 374.2115, found 374.2111. ^13^C{^1^H}-NMR (75 MHz, δ ppm/CDCl_3_): 165.5 (C), 163.2 (C), 137.4 (C), 135.8 (C), 131.4 (CH), 131.3 (C), 129.0 (CH), 129.0 (CH), 128.5 (CH), 128.5 (CH), 128.3 (CH), 128.3 (CH), 127.8 (CH), 127.8 (CH), 127.3 (CH), 127.2 (CH), 107.7 (C), 93.5 (CH), 46.2 (CH_2_), 45.7 (CH_2_), 41.1 (CH_2_), 36.2 (C), 30.0 (CH_3_), 29.8 (CH_3_), 13.9 (CH_3_). ^1^H-NMR (300 MHz, δ ppm/CDCl_3_): 7.38–7.19 (m, 10H), 6.37 (s, 1H), 5.65 (s, 1H), 5.32 (d, *J* = 15.4 Hz, 1H), 3.96 (d, *J* = 15.4 Hz, 1H), 2.46–2.33 (m, 4H), 1.90 (d, *J* = 1.2 Hz, 3H), 1.20 (s, 3H), 1.16 (s, 3H). 

**4d**: Following the general procedure, the reaction between α-chloro-cinnamaldehyde (144 mg, 0.86 mmol), benzylamine (94 μL, 0.86 mmol), and 5,5-dimethyl-2-diazo-cyclohexan-1,3-dione (144 mg, 0.86 mmol) at 140 °C for 15 min afforded the 1,3-oxazin-4-one **4d** (210 mg, 61%) as a colorless oil. Rf (Et_2_O/petroleum ether 4:6) = 0.35. MS (ESI+) *m*/*z*: 316 [M+Na]^+^, 332 [M+K]^+^. ^13^C{^1^H}-NMR (75 MHz, δ ppm/CDCl_3_): 164.8 (C), 161.9 (C), 136.5 (C), 132.9 (C), 129.5 (CH), 129.5 (CH), 129.1 (CH), 128.9 (CH), 128.7 (CH), 128.7 (CH), 128.3 (CH), 128.3 (CH), 128.0 (CH), 128.0 (CH), 127.6 (CH), 127.1 (C), 108.1 (C), 90.3 (CH), 46.2 (CH_2_), 46.1 (CH_2_), 41.0 (CH_2_), 36.3 (C), 29.9 (CH_3_), 29.1 (CH_3_). ^1^H-NMR (300 MHz, δ ppm/CDCl_3_): 7.58 (dd, *J* = 1.7, 7.3 Hz, 2H), 7.41–7.29 (m, 7H), 6.62 (s, 1H), 5.76 (s, 1H), 5.44 (d, *J* = 15.4 Hz, 1H), 3.99 (d, *J* = 15.4 Hz, 1H), 2.53–2.29 (m, 5H), 1.19 (s, 3H), 1.13 (s, 3H). 

**4f**: Following the general procedure, the reaction between crotonaldehyde (76 μL, 0.91 mmol), furfurylamine (81 μL, 0.91 mmol), and 5,5-dimethyl-2-diazo-cyclohexan-1,3-dione (152 mg, 0.91 mmol) at 140 °C for 15 min afforded the 1,3-oxazin-4-one **4f** (53 mg, 20%) as a colorless oil. Rf (Et_2_O/petroleum ether 2:8) = 0.41. HRMS (ESI+) *m/z*: [M+H]^+^ calcd for C_17_H_22_NO_3_^+^ 288.1594, found 288.1591. ^13^C{^1^H}-NMR (75 MHz, δ ppm/CDCl_3_): 165.9 (C), 162.5 (C), 151.0 (C), 142.0 (CH), 133.4 (CH), 124.4 (CH), 110.3 (CH), 108.1 (CH), 107.9 (C), 89.7 (CH), 46.1 (CH_2_), 41.1 (CH_2_), 38.2 (CH_2_), 36.2 (C), 30.0 (CH_3_), 29.9 (CH_3_), 17.5 (CH_3_). ^1^H NMR (300 MHz, δ ppm/CDCl_3_): 7.31–7.30 (m, 1H), 6.28–6.20 (m, 2H), 5.85–5.73 (m, 2H), 5.56 (d, *J* = 6.4 Hz, 1H), 4.87 (d, *J* = 15.7 Hz, 1H), 4.18 (d, *J* = 15.7 Hz, 1H), 2.37–2.27 (m, 4H), 1.73 (d, *J* = 6.4 Hz, 3H), 1.13 (s, 3H), 1.12 (s, 3H). 

**4g**: Following the general procedure, the reaction between cinnamaldehyde (126 μL, 1.0 mmol), benzylamine (109 μL, 1.0 mmol), and 2-diazo-cycloheptan-1,3-dione (152 mg, 1.0 mmol) at 160 °C for 15 min afforded the 1,3-oxazin-4-one **4g** (crude product that required no purification, 379 mg with purity >90% from the NMR analyses, quantitative) as a brown oil. HRMS (ESI+) *m/z*: [M+H]^+^ calcd for C_23_H_24_NO_2_^+^ 346.1802, found 346.1807. ^13^C{^1^H}-NMR (75 MHz, δ ppm/CDCl_3_): 163.2 (C), 160.7 (C), 137.2 (C), 135.3 (CH), 134.7 (C), 128.2 (CH), 128.2 (CH), 128.2 (CH), 128.2 (CH), 127.8 (CH), 127.4 (CH), 127.4 (CH), 126.9 (CH), 126.5 (CH), 126.5 (CH), 121.8 (CH), 106.5 (C), 86.8 (CH), 45.3 (CH_2_), 26.9 (CH_2_), 21.6 (CH_2_), 21.5 (CH_2_), 21.1 (CH_2_). ^1^H-NMR (300 MHz, δ ppm/CDCl_3_): 7.35–7.16 (m, 10 H), 6.56 (d, *J* = 15.9 Hz, 1H), 6.31 (dd, *J* = 7.5, 15.9 Hz, 1H), 5.61 (d, *J* = 7.5 Hz, 1H), 5.11 (d, *J* = 15.5 Hz, 1H), 4.25 (d, *J* = 15.5 Hz, 1H), 2.45 (br, 2H), 2.19 (br, 2H), 1.68 (br, 4H). 

**4h**: Following the general procedure, the reaction between cinnamaldehyde (126 μL, 1.0 mmol), *n*-propylamine (82 μL, 1.0 mmol), and 2-diazo-acetylacetone (126 mg, 1.0 mmol) at 140 °C for 5 min afforded the 1,3-oxazin-4-one **4h** (213 mg, 78%) as an orange oil. Rf (EtOAc/petroleum ether 2:8) = 0.27. MS (ESI+) *m/z*: 294 [M+Na]^+^. ^13^C{^1^H}-NMR (75 MHz, δ ppm/CDCl_3_): 163.5 (C), 158.0 (C), 135.0 (CH), 135.0 (CH), 128.4 (CH), 128.4 (CH), 128.4 (CH), 126.7 (CH), 126.7 (CH), 122.7 (CH), 105.0 (C), 87.0 (CH), 44.7 (CH_2_), 21.5 (CH_2_), 16.9 (CH_3_), 11.1 (CH_3_), 10.1 (CH_3_). ^1^H-NMR (300 MHz, δ ppm/CDCl_3_): 7.38–7.24 (m, 5H), 6.63 (d, *J* = 15.9 Hz, 1H), 6.33 (dd, *J* = 15.9, 6.6 Hz, 1H), 5.54 (d, *J* = 6.6 Hz, 1H), 3.65 (ddd, *J* = 13.9, 7.7, 7.1 Hz, 1H), 2.96 (ddd, *J* = 13.9, 7.7, 7.1 Hz, 1H), 1.87 (s, 3H), 1.76 (s, 3H), 1.62–1.49 (m, 2H), 0.86 (t, *J* = 7.5 Hz, 3H).

**iso-3h**: Following the general procedure, the reaction between cinnamaldehyde (151 μL, 1.2 mmol), *n*-propylamine (98 μL, 1.2 mmol), and 2-diazo-acetylacetone (151 mg, 1.2 mmol) at 200 °C for 15 min afforded the hydropyridin-2-one **iso-3h** (123 mg, 38%) as a yellow oil. Rf (EtOAc/pentane 1:10) = 0.62. HRMS (ESI+) *m/z*: [M+H]^+^ calcd for C_17_H_22_NO_2_^+^ 272.1645, found 272.1645. ^13^C{^1^H}-NMR (101 MHz, δ ppm/CDCl_3_): 208.1 (C), 170.0 (C), 138.4 (C), 129.3 (CH), 129.1 (2CH), 128.9 (2CH), 127.9 (CH), 108.0 (CH), 59.7 (C), 50.4 (CH), 48.7 (CH_2_), 30.0 (CH_3_), 22.6 (CH_3_), 22.0 (CH_2_), 11.4, (CH_3_). ^1^H-NMR (400 MHz, δ ppm/CDCl_3_): 7.31–7.24 (m, 3H), 7.15–7.13 (m, 2H), 6.29 (dd, *J* = 7.9, 0.9 Hz, 1H), 5.28 (dd, *J* = 8.0, 5.2 Hz, 1H), 3.65–3.50 (m, 2H), 3.46 (d, *J* = 5.2 Hz, 1H), 1.71 (q, *J* = 7.5 Hz, 2H), 1.55 (s, 3H), 1.50 (s, 3H), 1.0 (t, *J* = 7.5 Hz, 3H). The relative stereochemistry in **iso-3h** was determined by a NOESY experiment. 

**4i**: Following the general procedure, the reaction between cinnamaldehyde (126 μL, 1.0 mmol), allylamine (75 μL, 1.0 mmol), and methyl 2-diazo-acetylacetate (142 mg, 1.0 mmol) at 160 °C for 5 min afforded the 1,3-oxazin-4-one **4i** (124 mg, 43%) as a yellow oil. Rf (EtOAc/petroleum ether 2:8) = 0.37. MS (ESI+) *m/z*: 308 [M+Na]^+^. ^13^C{^1^H}-NMR (75 MHz, δ ppm/CDCl_3_): 170.6 (C), 167.9 (C), 139.0 (C), 132.3 (CH), 128.7 (CH), 128.7 (CH), 128.6 (CH), 128.0 (CH), 128.0 (CH), 127.4 (CH), 117.5 (CH_2_), 108.0 (CH), 54.7 (C), 51.6 (CH_3_), 49.9 (CH), 48.7 (CH_2_), 21.4 (CH_3_). ^1^H-NMR (300 MHz, δ ppm/CDCl_3_): 7.32–7.15 (m, 3H), 7.14–7.09 (m, 2H), 6.22 (dd, *J* = 7.8, 2.2 Hz, 1H), 5.93–5.77 (m, 1H), 5.33–5.21 (m, 3H), 4.26 (dd, *J* = 15.3, 5.6 Hz, 1H), 4,15 (dd, *J* = 15.3, 5.9 Hz, 1H), 3.66 (dd, *J* = 3.8, 2.1 Hz, 1H), 3.32 (s, 3H), 1.55 (s, 3H).

## 4. Conclusions

The formal aza-Diels–Alder cycloadditions of α-oxoketenes with 1-azadienes were examined in detail. Two distinct pericyclic products were obtained experimentally: 1,3-oxazin-4-ones of type **4** when the α-oxoketenes reacted as 4π partners (1-oxadienes) with the C=N bond of the 1-azadienes as 2π components, and hydropyridin-2-ones of type **3** when the 1-azadienes reacted as the 4π partners with the C=C bond of the ketene groups as the 2π components. Generally, it was shown experimentally that 1,3-oxazin-4-ones **4** are the kinetic products, and that hydropyridin-2-ones **3** can be obtained from reactions conducted at higher temperature, indicating some degree of thermodynamic control. However, some small variations in the structure of the α-oxoketenes or the 1-azadienes have led to sharp differences in the periselectivity, as well as inversion of the diastereoselectivity with an acyclic α-oxoketene. Computational modeling of these aza-Diels–Alder cycloadditions by DFT methods confirmed that the 1,3-oxazin-4-ones **4** are invariably the kinetic products, as well as that both diastereomers of hydropyridin-2-ones **3** are thermodynamically favored with, in all cases, the actual thermodynamic diastereomers computationally identified as **iso-3**. Experimentally, when possible, the reactions with cyclic α-oxoketenes have led exclusively to hydropyridin-2-ones **3**, not their diastereomers **iso-3**. In these cases, the computational model allowed identifying that formation of the thermodynamic product **iso-3** is actually kinetically impeded under the examined reaction conditions, notably by the formation of the α-oxoketenes cyclodimers. This situation is different for reactions with an acyclic α-oxoketene, which led experimentally to the thermodynamic product **iso-3**. In this case, the model shows that formation of **iso-3** is now kinetically allowed under the examined reaction conditions. This difference of behavior between cyclic and acyclic α-oxoketenes results from the existence of the additional and more stable *s-trans* conformation in acyclic α-oxoketenes. All in all, the factors governing periselectivity and diastereoselectivity in the aza-Diels–Alder cycloadditions of α-oxoketenes with 1-azadienes have been fully identified by a combined experimental and theoretical approach.

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
