# Peer review of "Periselectivity in the Aza-Diels–Alder Reaction of 1-Azadienes with α-Oxoketenes: A Combined Experimental and Theoretical Study"

_molecules, 2020, doi:10.3390/molecules25204811_

Round 1

Reviewer 1 Report

The paper is well written and interesting.

It is however unclear to me why in many instances (but not all - see e.g. Figure 5 and 6) the authors choose to calculate the deltaG of all the compounds at 298K, and whether it is these numbers they use for the calculation of the delta deltaG's at higher temperatures between the intermediates and transition states.

Please explain more carefully why this was done - should not the starting materials and intermediates, at the very least those in the rate determining step, be calculated at the reaction temperature too, in order to make a sensible prediction of the temperature dependence (as it is indeed done in Figs 5 and 6). The relative activation energy between possible paths also depends on the Gibbs free energy of the starting intermediate, and this appears not to be taken into account. That seems quite fundamental to the results and conclusions obtained.

In figure 1 - it appears the kinetically favored path would be a) here / whereas the thermodynamically favoured path would be b)/c). d) is invoked to explain the lack of formation of iso-3j, but from the numbers in Figure 1d) it actually appears as if there is a much larger activation energy for the dimerization than for a) ...  the energy of the starting material and product in d) is not given. I'm wondering about the reference point here (I suppose also the starting materials are taken as 0 ?).

If those remarks and their repercussions can be answered or addressed, I have no issue at all with this work being published in Molecules.

Author Response

The deltaG calculated at the reaction temperature for all kinetically meaningful TS, as well as those of the corresponding starting intermediates, are now shown in all Figures. The SI file was modified accordingly. Where appropriate, the text and numbers in the discussion were adjusted to reflect these complementary data, which does not change the conclusions.

The relative energies of the starting material and product in Figure 1d, Figure 2c, Figure 5d and Figure 6d are now provided; simple reaction arrows are now depicted instead of equilibrium arrows for homogeneity reasons. The cartesian coordinates and free Gibbs energies of dimer-1e, dimer-1b and dimer-1c were added to the SI file.

Reviewer 2 Report

The manuscript described the formal aza-Diels-Alder reaction of 1-azadienes with a-oxoketenes by Rodriguez and Coquerel et al. reaction afforded the unexpected product, however, the authors characterize these compounds well. In addition, for reveal the reaction mechanisms, the DFT calculation has been done and fully rationalized.

From these reasons, this manuscript could be published in Molecules without further revision.

Author Response

no change.

Reviewer 3 Report

In this manuscript (“Periselectivity in the aza-Diels–Alder reaction of 1-azadienes with alpha-oxoketenes: a combined experimental and theoreticalstudy”), the authors integrate the results discussed in a previous paper “Periselectivity in the aza-Diels−Alder Cycloaddition between α‑Oxoketenes and N‑(5-Pyrazolyl)imines: A Combined Experimental and Theoretical Study” (see ref. 5). Methodology of investigation, rationalization and conclusion are the same: periselectivity depends strongly on temperature, cinetic and termodynamic products are isolated and characterized, and the corresponding reaction paths are discussed.

The manuscript is well organized, both experimental and computational parts are exhaustive and results could be useful in synthesis. For this reason, I would like to recommend the publication after minor revision: 

1) I suggest the authors to read carefully the text and improve the English. Some sentences are too wide and too long. An example:

“Re-examination of the temperature dependence of the reaction outcome revealed that the other pericyclic product, namely the 1,3-oxazin-4-one 4a, is actually formed when the reaction is performed at 140 °C with a 3a/4a ratio determined at 2:1 after 15 min, which allowed isolating 3a in 38% yield and 4a in 21% yield; a similar reaction conducted at 150 °C for 30 min afforded only 3a in 22% isolated yield (3a/4a > 25:1).”

Moreover I would replace “with a 3a/4a ratio determined at 2:1”  with “with a 2:1 3a/4a ratio”

2) Figure 7: put the Energy value below the “LUMO-strans-1c” picture.

Author Response

1) the text was re-worked as suggested on a few occasions (see track-change version).

2) the energy value for the LUMO of strans-1c was added to Figure 7 and the average LUMO/LUMO+1 gap was recalculated.

Round 2

Reviewer 1 Report

I'm happy with the paper and the conclusions now, based on the computational data shown